# Anthropometric and Body Composition Changes during Pre-Season of Spanish Professional Female Soccer Players According to Playing Position

**DOI:** 10.3390/nu16162799

**Published:** 2024-08-22

**Authors:** Marta Ramírez-Munera, Raúl Arcusa, Francisco Javier López-Román, Desirée Victoria-Montesinos, Ana María García-Muñoz, Vicente Ávila-Gandía, Silvia Pérez-Piñero, Javier Marhuenda

**Affiliations:** 1Faculty of Pharmacy and Nutrition, Universidad Católica de Murcia (UCAM), Campus de los Jerónimos, Guadalupe, 30107 Murcia, Spain; marta.ramirez.nutricionista@gmail.com (M.R.-M.); dvictoria@ucam.edu (D.V.-M.); amgarcia13@ucam.edu (A.M.G.-M.); jmarhuenda@ucam.edu (J.M.); 2Faculty of Medicine, Universidad Católica de Murcia (UCAM), Campus de los Jerónimos, Guadalupe, 30107 Murcia, Spain; jlroman@ucam.edu (F.J.L.-R.); vavila@ucam.edu (V.Á.-G.); sperez2@ucam.edu (S.P.-P.)

**Keywords:** female soccer, anthropometry, bioimpedance, performance

## Abstract

Background: In professional soccer, body composition analysis is crucial to assess preparation and optimize performance. Different playing positions have different physical demands, which can lead to variations in body composition. However, there are few studies on women’s soccer that consider the playing position. This study aims to fill that gap by examining position-specific differences in anthropometric and body composition characteristics among Spanish professional female soccer players at the beginning and end of pre-season. Furthermore, it investigates the possible changes during the pre-season period between positions and correlates the data obtained from anthropometric equations with bioimpedance (BIA) measurements. Methods: Thirty-four female soccer players: 8 midfielders, 12 defenders, 11 forwards, and 3 goalkeepers (age: 23.06 ± 4.29 years, height: 164.15 ± 5.84 cm, weight: 58.39 ± 6.62 kg, and ∑6 skinfolds: 74.57 ± 18.48 mm) completed the study that lasted 4 weeks (pre-season) where they were measured anthropometrically and by bioimpedance twice. Results: Goalkeepers showed greater wingspan (176.60 ± 7.06 *p* < 0.05) compared to other positions. Regarding differences during pre-season, midfielders had the greatest decrease in ∑6 skinfolds compared to other positions (∆ −12.10 ± 5.69 *p* < 0.05). There was a correlation of % fat between Faulkner’s equation and BIA (Pearson’s *r* = 0.817). Conclusions: It seems that there are no significant differences in terms of positions and body composition, except for the wingspan and ankle diameter. During pre-season, midfielders are the ones who improve their body composition the greatest. The anthropometric equation for body fat that shows the highest correlation with BIA is Faulkner’s equation, followed by Durnin’s equation.

## 1. Introduction

In recent years, there has been a significant surge in interest towards women’s soccer [1]. According to data from the Fédération Internationale de Football Association (FIFA), the number of women’s clubs in 2023 was around 55,600, 59% of which were located in Europe [2]. In Spain, the number of federation licenses for women’s football in recent years has increased considerably, exceeding the figure of 87,000 in 2022, with a growth rate of 30.8% from 2021 to 2022 [3]. Moreover, research about women’s soccer has also increased [4], although it still remains limited for female players compared to their male counterparts [5].

Anthropometric studies allow the estimation of body composition, the study of morphology, dimensions and proportionality in relation to sports performance and nutrition [6]. One aim of analyzing body composition is to identify and measure different body compartments. In professional soccer, these assessments are used alongside fitness tests to assess readiness for competition and to track how training and diet changes affect body composition [7], and in this way, optimize body composition in order to achieve an improvement in sporting performance throughout the season.

Different positions can be distinguished in a soccer team such as forwards, midfielders, defenders, and goalkeepers. For example, forwards and wingers (type of defender who plays on the flank and carries out numerous sprints while playing) often make intense bursts of speed and agility, requiring explosive power and rapid acceleration to penetrate opposing defenses. Midfielders, meanwhile, often cover extensive distances during a match, requiring exceptional stamina and endurance to contribute both defensively and offensively. Defenders focus on strength, agility, and spatial awareness to disrupt opposing attackers and initiate build-up play. Goalkeepers require agility, reflexes and explosive movement to make decisive saves [8]. These distinct roles may contribute to differences in players’ body composition and physical attributes due to the specialized movements and performance expectations associated with their positions [9].

In Spain, there is limited research that provides some information on body composition or anthropometric values in women’s soccer [10,11,12,13,14,15,16,17,18]. Most of the data are about body composition without absolute anthropometric data, or with anthropometric data limited to the measurement of skinfolds. In addition, there are only four studies that compare values between playing positions: one of them is performed on adolescent players [13], another one is performed on elite players (however, the absolute differences for elite players were not available as values were combined with non-elite players) [11], and finally, to our knowledge, there are only two research studies with body composition data of Spanish professional players comparing between playing positions, published in 2009 [10] and more recently in 2024 [18]. Therefore, there is a clear need for updated data of Spanish professional players by playing position. In the same way, at the international level, positional differences of elite female players have been assessed in a small number of studies [1].

The pre-season period in the Spanish Women’s Football League lasts between 5 and 7 weeks, depending on each team’s schedule. This period is seen as the final opportunity for coaches to enhance the technical, tactical, and physical capabilities, including body composition, of their teams before starting the competitive period [19], usually by systematically increasing the training load and providing individualized dietary regimes [20]. To the best of our knowledge, this research appears to be the first to analyze how anthropometric and body composition parameters change across different playing positions in female soccer players during pre-season. Previous studies have explored changes in these parameters throughout the season, but they typically focused on the entire team without considering positional differences [16,19,21,22].

Due to the lack of scientific literature on women’s football and anthropometric data, the aim of the present research is to contribute to identify position-specific differences in the body composition characteristics of Spanish professional female football players at the beginning and end of pre-season and, as a secondary aim, to examine the correlation of different anthropometric equations with bioimpedance (BIA).

## 2. Materials and Methods

### 2.1. Study Design

A prospective study was carried out during pre-season with the aim of evaluating changes in anthropometric and BIA variables in professional female soccer players. All determinations were performed on all players at two different times: at the beginning and at the end of pre-season (4 weeks). The protocol was approved by the Institutional Review Committee of the Catholic University San Antonio of Murcia (UCAM) (code CE052204). This research was carried out following the Standards of Good Clinical Practice and was conducted according to the Declaration of Helsinki. The clinical trial was registered at www.clinicaltrials.gov (identifier NCT05525871). The study was carried out at the Department of Exercise Physiology of the Universidad Católica de Murcia (UCAM). The current European legislation on the protection of personal data (Regulation (EU)2016/679) was complied with.

### 2.2. Participants

The study was carried out in collaboration with two soccer teams from the first and second Spanish Women’s Football League. In total, 34 female players (age: 23.06 ± 4.29 years) were distributed according to the usual playing positions: 8 midfielders (age: 25.00 ± 3.46 years), 12 defenders (age: 23.08 ± 3.15 years), 11 forwards (age: 22.27 ± 5.64 years), and 3 goalkeepers (age: 20.67 ± 4.50 years).

The inclusion criteria were as follows: (a) be a healthy subject with medical authorization for the practice of a federated sport and (b) belong to a team of the first or second division of the Spanish Women’s League. The exclusion criteria for the study were as follows: (a) changing teams during pre-season and (b) a history of drug, alcohol or other substance abuse, or other factors limiting their ability to cooperate during the study. All players were previously informed of the objectives and method of the research, signing informed consent forms before starting the research. In the case of those players who were minors, it was their parents who signed the consent.

### 2.3. Anthropometric Measurements and Bioimpedance Analysis

The determinations were measured as follows: four basic measurements (body mass, height, sitting height, and arm span), eight skinfolds (biceps, triceps, subscapular, suprailiac, supraspinal, abdominal, front thigh, and medial calf), six breadths (humerus, femur, biacromial, bi-styloid, bi-iliocristal, and bimalleolar), and six perimeters (arm relaxed, arm flexed and tensed, waist, hip, thigh, and calf).

Body mass measurement and BIA analysis were determined with Tanita BC 420 S MA Class III, tetra polar system and single frequency (50 kHz). Height was measured using a scale with a stadiometer (Seca, Hamburg, Germany) to the nearest 0.1 cm; a SECA 217 detachable portable stadiometer (SECA, Germany) to the nearest 0.1 cm and anthropometry box were used to measure sitting height; a CESCORF inextensible metal tape (CESCORF, Porto Alegre, Brazil) was used to measure perimeters and arm span; a Harpenden skinfold caliper to the nearest 0.1 mm was used to measure skinfolds; and large and small sliding calipers (CESCORF, Porto Alegre, Brazil) were used to measure breadths. All anthropometric measurements were measured two or three times by an anthropometrist who was level 2 accredited by International Society for the advancement of the Kinanthropometry (ISAK), following the procedures established by the ISAK [23].

Body composition was determined using the equations described in the consensus document of the Spanish Group of Kinanthropometry of the Spanish Federation of Sports Medicine [6], following the four-component model (muscle mass (MM), fat mass (FM), bone mass (BM), and residual mass (RM)). The following equations were used: the Withers [24], Slaugther [25], Carter [26], Faulkner [27], Durnin [28], and Jackson and Pollock [29] equations to calculate FM expressed in percentage (%) and kilograms (kg); the Poortmans [30] and Lee [31] equations to calculate MM expressed in kg and %; and Rocha’s equation [32] to calculate BM expressed in kg and %. The sum of 3, 6, and 8 skinfold measurements were also calculated. The sum of 8 skinfolds includes all measured skinfolds, the sum of 6 skinfolds excludes the bicipital and suprailiac skinfolds, while the sum of 3 skinfolds are located near the abdominal area (suprailiac, supraspinal, and abdominal).

The Heath–Carter [33] method was used to estimate anthropometric somatotypes (endomorphy, mesomorphy, and ectomorphy).

### 2.4. Statistical Analyses

SPSS Statistics 27 (SPSS, Inc., Chicago, IL, USA) was used for statistical analysis. Descriptive statistics were calculated and used to describe the anthropometric and BIA characteristics of the players.

The statistics were reported as mean ± standard deviation (SD). One-way analysis of variance (ANOVA) was used to determine differences between the anthropometric measures and BIA analysis among the four playing positions. Tukey’s post hoc test was used to determine which variables differed significantly. The level of statistical significance was set at *p* ≤ 0.05. Pearson’s correlation coefficient (*r*) was used to determine the possible correlation between body composition obtained by anthropometric equations and body composition obtained by BIA.

## 3. Results and Discussion

Table 1 shows the baseline demographic characteristics of the players; for this reason, no *p*-values have been included. Table 2 includes the *p*-values at baseline since the deviation was residual as they were bone diameters. Furthermore, from Table 3 onwards, it was only considered to include the *p*-value of the variation during pre-season (end of pre-season—baseline) because no significant changes were found in the values at the beginning and the end of pre-season regarding playing positions. According to these findings, the similarity between playing positions is consistent with some previous studies in the literature [10,34,35,36,37]. Other studies did report significant differences between positions, but it is important to mention the diversity of the study populations, because the studies reporting differences have not studied populations composed exclusively of elite or professional players. One study involves a mix of Spanish elite and non-elite players [11], while another study includes Chilean professional and collegiate players [38]. Additionally, there are studies where all players are from South African semi-professional or collegiate clubs [39], and another study, despite the team being national, specifies that the players compete in semi-professional teams in South Africa [40]. In studies on professional or elite female players, minor differences were reported [41,42,43].

Table 2 shows the descriptive statistics for the different body dimensions, such as standing and sitting height, wingspan, bone diameters, bone mass, and age by both playing and total positions. These variables were only considered to be measured at baseline, as bone diameters and lengths do not change in such a short period of time (4 weeks).

The variation of these body dimensions showed statistically significant differences (*p* < 0.05) in only two variables (wingspan and ankle diameter). Both variables showed differences between goalkeepers and the rest of the positions. No differences were observed between the rest of the positions (*p* > 0.05). Regarding wingspan, it was observed that goalkeepers were taller than the rest of the positions (goalkeepers vs. midfielders: ∆ = 15.30 cm, *p* = 0.004) (goalkeepers vs. defenders: ∆ = 13.11 cm, *p* = 0.012) (goalkeepers vs. forwards: ∆ = 12.26 cm, *p* = 0.021). That high wingspan diameter in goalkeepers can be explained by the need to effectively block shots and defend the goal against the opposing team’s attempts. Comparing with other studies, these findings are in line with results from studies on Spanish female players aged 16 and 18 [13]. That study also showed the greatest wingspan in goalkeepers, but it was not specified if it was statistically significant compared with the rest of the playing positions. Nonetheless, a study of Ecuadorian female players [44] reported similar values for wingspan between playing positions. It is important to note that in this study, players were divided into eight different categories with only two goalkeepers included, which may limit the robustness of these findings. On the contrary to wingspan, ankle diameter showed lower values in goalkeepers (5.80 ± 0.89 cm) compared to midfielders (6.93 ± 0.38 cm), defenders (6.70 ± 0.28 cm), and forwards (6.66 ± 0.56 cm), with a *p*-value of 0.012. This contrasts with a study on South African sub-elite female players [39], which reported ankle diameters of 6.4 ± 0.4 cm for forwards, 6.4 ± 0.3 cm for midfielders, 6.6 ± 0.4 cm for defenders, and 6.7 ± 0.3 cm for goalkeepers, with a *p*-value of 0.034. Although the South African study found statistically significant differences, the practical relevance was unclear due to wide confidence intervals and small effect sizes. The discrepancies between the two studies could be attributed to differences between categories. The present study involves professional players who have different training intensities and physical demands compared to the sub-elite players [45]. Additionally, ethnicity [7], genetic and environmental factors [46], as well as potential variations in measurement methods, might contribute to these differences. It is important to highlight that there is a limited number of studies that have examined or considered both wingspan and ankle diameter in female soccer players in different playing positions.

Regarding height, goalkeepers were shown to be the tallest players. That fact agrees with the observations reported for Tunisian [41], Croatian [47,48], and Chilean [37] first division players, sub-elite and national team South African players, and Chilean players, including both national team members [42] and professional and elite college players [38]. Other studies reported slight differences. For example, a study revealed that the Norwegian first and second division goalkeepers [34] were, along with the defenders, the tallest players on the team. Similarly, in the national team of Montenegro [35], the tallest players were again the goalkeepers, but in this case, their height was similar to that of the forwards. The height reported in the present study for goalkeepers (171.73 ± 8.39) is higher that that observed in other studies [34,37,38,39,40,41], including the study conducted in 2009 on Spanish first division players (1.64 ± 0.6) which reported the defenders as the tallest players [10], but is similar to a study in Croatian first division players (172.5) [47] (172.5 ± 4.6) [48] and the national teams of Chile (172.5 ± 6.6) [42] and Montenegro (170.5 ± 4.2) [35].

Continuing with height, but analyzing the lowest values, midfielders were shown to be the players with the shortest heights. This finding is concordant with observations in Danish [49] and Croatian [47] first division players, as well as sub-elite South African players [39]. It is also worth mentioning that in the national team of Montenegro [35], midfielders were the shortest, along with defenders. However, other studies reported the short height for forwards [34,36,38,40,42,43,48], leading to a wide distribution of height within different playing positions. To understand these differences observed in the different studies, it would be interesting to consider both the playing style of the different teams or divisions, as well as the criteria of the researchers to consider a player within a specific category of playing position. For example, it is not the same to consider a skilled player who plays on the wings (that is usually a small player) as a lateral midfielder or as a winger, since in the first case she would be defined as a midfielder, and in the second, as a forward.

**Table 2 nutrients-16-02799-t002:** Descriptive statistics for the different body dimensions of the players: mean and standard deviations (mean ± SD).

Variable	Position	Value	*p*-Value
Age	MF	25.00 ± 3.46	0.415
	DF	23.08 ± 3.15	
	FW	22.27 ± 5.64	
	GK	20.67 ± 4.51	
	Total	23.06 ± 4.29	
Height (cm)	MF	161.85 ± 5.48	0.089
	DF	164.11 ± 4.37	
	FW	163.79 ± 5.89	
	GK	171.73 ± 8.39	
	Total	164.15 ± 5.84	
Sitting height (cm)	MF	161.30 ± 5.78	0.506
	DF	163.49 ± 5.35	
	FW	164.34 ± 6.80	
	GK	176.60 ± 7.06	
	Total	164.41 ± 7.06	
Wingspan (cm)	MF	161.30 ± 5.78	0.008 *
	DF	163.49 ± 5.35	
	FW	164.34 ± 6.80	
	GK	176.60 ± 7.06 *	
	Total	164.41 ± 7.06	
Humerus (cm)	MF	6.31 ± 0.26	0.372
	DF	6.28 ± 0.25	
	FW	6.26 ± 0.29	
	GK	6.60 ± 0.61	
	Total	6.31 ± 0.30	
Femur (cm)	MF	8.94 ± 0.46	0.617
	DF	8.96 ± 0.24	
	FW	8.87 ± 0.30	
	GK	9.23 ± 1.01	
	Total	8.95 ± 0.41	
Biacromial (cm)	MF	34.84 ± 1.78	0.146
	DF	36.22 ± 1.50	
	FW	36.14 ± 2.43	
	GK	37.83 ± 1.75	
	Total	36.01 ± 2.01	
Wrist (bistyloid) (cm)	MF	5.03 ± 0.212	0.474
	DF	4.98 ± 0.18	
	FW	5.01 ± 0.25	
	GK	5.20 ± 0.27	
	Total	5.04 ± 0.22	
Bi-iliocristal (cm)	MF	26.71 ± 1.10	0.215
	DF	26.73 ± 1.03	
	FW	26.32 ± 1.63	
	GK	28.50 ± 3.50	
	Total	26.75 ± 1.58	
Ankle (bimalleolar) (cm)	MF	6.93 ± 0.38	0.012 *
	DF	6.70 ± 0.28	
	FW	6.66 ± 0.56	
	GK	5.80 ± 0.89 *	
	Total	6.66 ± 0.53	
Bone mass (kg) (Rocha)	MF	9.10 ± 0.74	0.134
	DF	9.25 ± 0.63	
	FW	9.27 ± 0.73	
	GK	10.45 ± 1.96	
	Total	9.33 ± 0.88	
%Bone mass (Rocha)	MF	15.81 ± 1.26	0.159
	DF	16.35 ± 1.54	
	FW	15.83 ± 0.86	
	GK	16.25 ± 0.46	
	Total	16.04 ± 1.20	

MF: midfielders; DF: defenders; FW: forwards; GK: goalkeepers. * Significant differences between positions at baseline (*p* < 0.05).

Table 3 shows the descriptive statistics at the beginning and end of pre-season, as well as their evolution for body mass in kg, BMI, and the different perimeters. In this research, the heaviest players were goalkeepers. This finding coincides with observations in Spanish [10,18], Chilean [37], Tunisian [41], and Croatian [47,48] first division players. This is also consistent with observations made in the national teams of Montenegro [35], Chile [42], and South Africa [40], as well as in Chilean professional and elite college players [38] and South African sub-elite players [39]. In turn, a recent review of the literature reveals that there is an almost equal number of studies reporting that forwards are the lightest [34,36,38,42,43] as those reporting that midfielders [18,35,39,41,47,48,49] or defenders [10,35,37,40,48] are the lightest. Consequently, the present research neither contradicts nor supports the existing studies on this matter, as there is no clear predominance of one position being the lightest. In terms of evolution during pre-season, the most noticeable fact was the decrease observed in the relaxed arm perimeter (*p* < 0.05), showing a reduction in all the positions except for goalkeepers. Comparing playing positions, goalkeepers and midfielders were the only ones that showed statistically significant differences (*p* = 0.017).

**Table 3 nutrients-16-02799-t003:** Descriptive statistics at baseline and at the end of pre-season, and variation of body mass, BMI, and perimeters: mean and standard deviations (mean ± SD).

Variable	Position	Baseline	Final	∆ Final—Baseline	*p*-Value
Body mass (kg)	MF	57.91 ± 6.43	57.18 ± 6.98	−0.74 ± 1.40	0.257
	DF	56.83 ± 4.25	57.03 ± 4.31	0.21 ± 1.37	
	FW	58.77 ± 6.53	58.51 ± 6.70	−0.26 ± 0.99	
	GK	64.53 ± 13.90	65.33 ± 12.14	0.80 ± 1.76	
	Total	58.39 ± 6.62	58.28 ± 6.67	−0.12 ± 1.32	
BMI (kg/m^2^)	MF	22.07 ± 1.90	21.78 ± 2.01	−0.29 ± 0.54	
	DF	21.12 ± 1.76	21.20 ± 1.71	0.07 ± 0.51	
	FW	21.86 ± 1.61	21.76 ± 1.69	−0.09 ± 0.35	0.228
	GK	21.68 ± 2.44	21.99 ± 1.84	0.31 ± 0.59	
	Total	21.63 ± 1.76	21.59 ± 1.73	−0.04 ± 0.49	
Relaxed arm (cm)	MF	27.39 ± 2.14	26.90 ± 2.35	−0.49 ± 0.40 *	0.027 *
	DF	26.23 ± 1.84	26.15 ± 1.71	−0.08 ± 0.64	
	FW	26.94 ± 1.37	26.72 ± 1.52	−0.22 ± 0.44	
	GK	27.13 ± 2.20	27.77 ± 1.63	0.63 ± 0.57	
	Total	26.81 ± 1.78	26.65 ± 1.80	−0.16 ± 0.58	
Contracted arm (cm)	MF	28.91 ± 2.29	28.03 ± 2.06	−0.87 ± 0.81	
	DF	27.29 ± 1.65	27.21 ± 1.61	−0.07 ± 0.53	
	FW	28.01 ± 1.35	27.83 ± 1.26	−0.18 ± 0.74	0.105
	GK	28.83 ± 2.30	28.86 ± 1.15	0.03 ± 1.32	
	Total	28.04 ± 1.83	27.75 ± 1.60	−0.28 ± 0.78	
Waist (cm)	MF	69.05 ± 3.38	68.03 ± 4.04	−1.01 ± 1.13	
	DF	70.42 ± 4.26	69.94 ± 4.22	−0.48 ± 1.29	
	FW	70.41 ± 3.90	69.74 ± 3.91	−0.67 ± 1.44	0.525
	GK	71.40 ± 7.45	71.70 ± 6.15	0.30 ± 1.51	
	Total	70.18 ± 4.11	69.58 ± 4.17	−0.60 ± 1.31	
Hip (cm)	MF	97.46 ± 4.70	96.78 ± 5.10	−0.67 ± 2.09	
	DF	95.99 ± 4.24	97.03 ± 3.81	1.04 ± 2.92	
	FW	97.58 ± 4.02	97.30 ± 4.58	−0.27 ± 1.24	0.356
	GK	102.33 ± 11.00	102.23 ± 9.09	−0.10 ± 2.47	
	Total	97.41 ± 5.08	97.52 ± 4.90	0.11 ± 2.26	
Thigh (cm)	MF	51.03 ± 2.93	50.42 ± 2.67	−0.61 ± 0.87	
	DF	49.73 ± 3.30	50.01 ± 2.91	0.28 ± 1.05	
	FW	51.60 ± 2.41	51.44 ± 2.45	−0.16 ± 0.63	0.106
	GK	50.20 ± 5.02	50.73 ± 4.61	0.53 ± 0.68	
	Total	50.68 ± 3.06	50.63 ± 2.79	−0.05 ± 0.91	
Calf (cm)	MF	36.13 ± 1.64	36.06 ± 1.98	−0.07 ± 0.60	
	DF	34.40 ± 1.69	34.38 ± 1.61	−0.01 ± 0.88	
	FW	35.46 ± 1.46	35.60 ± 1.35	0.13 ± 0.25	0.905
	GK	36.50 ± 2.95	36.53 ± 2.30	0.03 ± 0.66	
	Total	35.33 ± 1.81	35.36 ± 1.78	0.02 ± 0.62	

MF: midfielders; DF: defenders; FW: forwards; GK: goalkeepers. * Significant differences between positions in pre-season variation (*p* < 0.05).

Regarding the type of skinfold and the position on the field, Table 4 shows a significant reduction in the triceps, front thigh, and medial calf. Midfielders exhibited the lowest values, with a significant decrease compared to defenders, especially in the triceps (*p* = 0.031), front thigh (*p* = 0.003), and medial calf (*p* = 0.005). Compared to other positions, midfielders showed a significant reduction in the medial calf compared to forwards (*p* = 0.025) and in the triceps compared to goalkeepers (*p* = 0.041). In this research, it is observed that the skinfolds with the highest millimeters are the front thigh, suprailiac, and triceps, followed by the abdominal and medial calf. The skinfolds with the lowest measurements are found in the biceps, subscapular, and supraspinal regions. This is consistent with observations from a study involving more than 700 female athletes from various disciplines, which shows that, aside from having higher skinfold measurements than their male counterparts, fat accumulation tends to occur more in the lower body and trunk areas [50].

Regarding the different sums of skinfolds, midfielders were also the position that showed the most pronounced decreases. The sum of the 3 skinfolds showed significant differences between positions (*p* = 0.041). The sum of 6 skinfolds indicated differences between midfielders and defenders (*p* = 0.001) and between midfielders and forwards (*p* = 0.030). Finally, the sum of 8 skinfolds (which includes all the measured skinfolds) revealed significant differences between midfielders and defenders (*p* = 0.004). Continuing with the sum of 6 skinfolds, goalkeepers were shown to be the players with the highest values, and forwards shown to be the players with the lowest values. A possible explanation for this is that goalkeepers cover less distance during the game and thus use less energy, while other players, such as forwards, are generally lighter to be able to cover more ground during the match. This finding about goalkeepers is concordant with observations in South African sub-elite players [39,40] and Chilean [38] professional and elite collegiate players. However, this differs from the observations in the Chilean first division [37] and national team [42] goalkeepers. In these studies, the players with the highest sum of 6 skinfolds were central defenders and defenders, respectively. In terms of values, the sum of 6 skinfolds at the end of pre-season for goalkeepers in this research was lower (78.57 ± 19.45) than those found in sub-elite South African [39,40] (89 ± 5.23) (125.6 ± 45.9) and professional and elite collegiate level Chilean players [38] (104.2 ± 36.2), but higher than those found in Chilean first division [37] (62.5 ± 22.0) and national team [42] (67.78 ± 13.3) players. In summary, the results of the present research for the sum of 6 skinfolds across positions are lower than those observed in semi-professional or sub-elite players, but higher than those observed in players from national teams, which can be understood due to the difference in levels between the players from the different studies.

**Table 4 nutrients-16-02799-t004:** Descriptive statistics at baseline and at the end of pre-season, and variation of skinfolds and sum of skinfolds: mean and standard deviations (mean ± SD).

Variable	Position	Baseline	Final	Final—Baseline	*p*-Value
Triceps (mm)	MF	15.19 ± 5.01	13.01 ± 4.06	−2.18 ± 1.45 *	0.017 *
	DF	12.86 ± 2.85	12.23 ± 2.75	−0.63 ± 0.88	
	FW	13.06 ± 3.13	12.19 ± 2.64	−0.86 ± 1.28	
	GK	14.17 ± 5.96	14.20 ± 5.38	0.03 ± 0.67	
	Total	13.59 ± 3.75	12.58 ± 3.20	−1.01 ± 1.31	
Subscapular (mm)	MF	8.51 ± 1.95	7.63 ± 1.81	−0.89 ± 0.56	0.206
	DF	8.19 ± 1.36	8.03 ± 1.21	−0.16 ± 1.01	
	FW	8.07 ± 1.34	7.86 ± 1.63	−0.22 ± 0.90	
	GK	9.07 ± 2.34	9.17 ± 2.20	0.10 ± 0.36	
	Total	8.31 ± 1.56	7.98 ± 1.56	−0.33 ± 0.88	
Biceps (mm)	MF	6.28 ± 3.47	5.73 ± 3.32	−0.55 ± 0.71	0.659
	DF	4.75 ± 1.35	4.60 ± 1.28	−0.15 ± 0.51	
	FW	5.27 ± 2.10	5.13 ± 1.98	−0.15 ± 1.33	
	GK	7.03 ± 4.30	7.17 ± 3.68	0.13 ± 0.81	
	Total	5.48 ± 2.49	5.26 ± 2.33	−0.22 ± 0.90	
Suprailiac (mm)	MF	14.90 ± 6.49	11.68 ± 4.96	−3.23 ± 3.08	0.066
	DF	13.24 ± 3.68	12.77 ± 2.99	−0.48 ± 2.53	
	FW	14.43 ± 3.87	11.81 ± 3.50	−2.62 ± 1.98	
	GK	15.27 ± 5.81	15.23 ± 2.84	−0.03 ± 3.71	
	Total	14.19 ± 4.54	12.42 ± 3.66	−1.78 ± 2.79	
Supraspinal (mm)	MF	7.66 ± 3.21	6.89 ± 3.20	−0.78 ± 1.09	0.826
	DF	7.36 ± 2.64	6.87 ± 1.77	−0.49 ± 1.29	
	FW	7.42 ± 1.77	6.93 ± 2.00	−0.49 ± 0.99	
	GK	8.60 ± 3.12	8.53 ± 2.44	−0.07 ± 1.07	
	Total	7.56 ± 2.48	7.04 ± 2.24	−0.52 ± 1.10	
Abdominal (mm)	MF	12.84 ± 4.99	10.93 ± 4.56	−1.91 ± 1.16	0.123
	DF	12.15 ± 2.70	11.26 ± 3.35	−0.89 ± 1.70	
	FW	12.79 ± 3.77	11.17 ± 3.00	−1.62 ± 1.22	
	GK	13.03 ± 1.82	13.23 ± 3.35	0.20 ± 1.68	
	Total	12.60 ± 3.51	11.33 ± 3.45	−1.27 ± 1.50	
Front thigh (mm)	MF	25.50 ± 9.07	21.00 ± 6.71	−4.50 ± 3.51 *	0.004 *
	DF	19.13 ± 4.19	18.73 ± 4.41	−0.40 ± 1.85	
	FW	19.62 ± 4.09	17.69 ± 3.44	−1.93 ± 1.30	
	GK	24.33 ± 8.10	20.67 ± 4.94	−3.67 ± 3.17	
	Total	21.24 ± 6.33	19.10 ± 4.79	−2.15 ± 2.75	
Medial calf (mm)	MF	13.14 ± 5.64	11.29 ± 4.51	−1.85 ± 1.65 *	0.003 *
	DF	10.08 ± 3.96	10.00 ± 3.84	−0.08 ± 0.50	
	FW	10.34 ± 4.18	9.97 ± 4.19	−0.36 ± 0.44	
	GK	14.50 ± 5.46	12.77 ± 3.25	−1.73 ± 2.20	
	Total	11.27 ± 4.66	10.54 ± 4.00	−0.735 ± 1.27	
Sum of Skinfolds					
∑3 skinfolds	MF	35.40 ± 13.33	29.49 ± 12.13	−5.91 ± 3.82	0.041 *
	DF	32.75 ± 7.84	30.89 ± 7.08	−1.86 ± 4.67	
	FW	34.64 ± 9.00	29.91 ± 8.10	−4.73 ± 2.61	
	GK	36.90 ± 10.41	37.00 ± 8.42	0.10 ± 3.16	
	Total	34.35 ± 9.54	30.78 ± 8.73	−3.57 ± 4.14	
∑6 skinfolds	MF	82.84 ± 26.24	70.74 ± 23.06	−12.10 ± 5.69 *	0.002 *
	DF	69.77 ± 12.24	67.12 ± 12.28	−2.65 ± 5.84	
	FW	71.29 ± 15.39	65.81 ± 14.45	−5.48 ± 1.80	
	GK	83.70 ± 24.85	78.57 ± 19.45	−5.13 ± 5.75	
	Total	74.57 ± 18.48	68.56 ± 16.21	−6.01 ± 5.87	
∑8 skinfolds	MF	104.01 ± 35.38	88.14 ± 30.58	−15.88 ± 9.08 *	0.007 *
	DF	87.76 ± 15.31	84.48 ± 14.92	−3.28 ± 8.08	
	FW	90.99 ± 20.04	82.75 ± 19.14	−8.25 ± 3.56	
	GK	106.00 ± 33.83	100.97 ± 24.39	−5.03 ± 9.88	
	Total	94.24 ± 24.27	86.24 ± 21.11	−8.00 ± 8.54	

MF: midfielders; DF: defenders; FW: forwards; GK: goalkeepers. * Significant differences between positions in pre-season variation (*p* < 0.05).

Table 5 and Table 6 show the descriptive statistics at the beginning and end of pre-season, as well as their evolution for absolute and relative values of FM, respectively, showing a general decrease. However, the reduction observed for goalkeepers was only related to equations that only consider folds in the full body, but not for those that only consider the skin folds of the upper body exclusively (Faulkner and Durnin).

Table 5 displays the kg of fat obtained by the different equations (*p* < 0.05). In all the equations, the lowest values were observed for midfielders, especially compared with defenders—Withers (*p* = 0.013), Slaughter (*p* = 0.010), Carter (*p* = 0.007), and Jackson and Pollock (*p* = 0.010)—and versus goalkeepers, with Faulkner (*p* = 0.030). In contrast, Durnin’s equation also revealed significant differences between positions (*p* = 0.036). Despite the specific positions not being revealed, the two positions that showed the greatest variation in this parameter were the midfielders (−1.18 ± 0.61 kg) and goalkeepers (0.25 ± 0.99 kg).

The % fat followed a similar trend as that observed for the kg of fat. Table 6 shows the lowest values in midfielders and significant decreases were observed versus defenders, Withers (*p* = 0.003), Slaughter (*p* = 0.003), Carter (*p* = 0.001), Faulkner (*p* = 0.038), and Jackson and Pollock (*p* = 0.003); versus forwards, Withers (*p* = 0.023), Slaughter (*p* = 0.015), and Carter (*p* = 0.030); and versus goalkeepers, Withers (*p* = 0.036), Faulkner (*p* = 0.016) and Durnin (*p* = 0.030). It is notable that positions with significant differences in skinfolds, such as defenders, lead to a minor improvement potential. Similarly, for the sum of 6 and 8 skinfolds, defenders started with the lowest values and, therefore, lower fat values while goalkeepers had the highest, followed by midfielders. This trend could explain the significant difference in progression between midfielders and defenders. Defenders had the lowest values at the beginning of pre-season, followed by forwards, midfielders, and goalkeepers. The differences observed in midfielders compared to defenders could be attributed to the initial differences in body composition. Similarly, while midfielders and goalkeepers often had similar baseline values, midfielders consistently demonstrated greater improvements. This difference in improvement might be linked to the specific training regimens for each position. This situation can also be observed later with other data such as somatotype or BIA values.

The lowest fat values were obtained using Carter’s equation, followed by Faulkner, Withers, Jackson and Pollock, and Slaughter, with the highest fat values provided by Durnin’s equation. Interpreting results requires understanding which skinfolds are considered in the anthropometric equations estimating body fat. For example, Carter uses an equation that includes the sum of 6 skinfolds. In contrast, other equations, such as Durnin and Faulkner, only consider upper body skinfolds. This can lead to an overestimation or underestimation of body fat percentage, as the lower body, a common site for fat accumulation in female athletes, is not accounted for [50]. Comparing with other studies, one of the limitations is the diversity of equations used to estimate body fat percentage. Other studies have used equations such as Sloan and Siri [47], Yuhasz [10], Jackson and Pollock [48], Kerr [38], Durnin [41], ∑6 skinfolds [37,42], Withers [39], and Eston 2005 [43].

**Table 5 nutrients-16-02799-t005:** Descriptive statistics at baseline and at the end of pre-season and variation of kg of FM calculated with different equations adapted for female athletes: mean and standard deviations (mean ± SD).

Variable	Position	Baseline	Final	∆ Final—Baseline	*p*-Value
Fat (kg) (Withers)	MF	10.74 ± 3.53	9.53 ± 3.62	−1.21 ± 0.58 *	0.015 *
	DF	9.48 ± 1.99	9.24 ± 1.94	−0.24 ± 0.82	
	FW	9.93 ± 2.58	9.46 ± 2.67	−0.47 ± 0.26	
	GK	12.60 ± 5.96	12.43 ± 4.91	−0.17 ± 1.06	
	Total	10.20 ± 2.99	9.66 ± 2.90	−0.54 ± 0.73	
Fat (kg) (Slaughter)	MF	13.13 ± 4.57	11.58 ± 3.88	−1.56 ± 1.27 *	
	DF	10.86 ± 2.18	10.66 ± 2.29	−0.20 ± 0.61	
	FW	11.49 ± 3.12	11.00 ± 2.97	−0.49 ± 0.41	0.015 *
	GK	15.14 ± 7.90	14.45 ± 6.21	−0.69 ± 1.69	
	Total	11.98 ± 3.81	11.32 ± 3.32	−0.66 ± 0.98	
Fat (kg) (Carter)	MF	9.64 ± 3.15	8.45 ± 2.82	−1.19 ± 0.68 *	0.013 *
	DF	8.20 ± 1.41	8.00 ± 1.45	−0.20 ± 0.70	
	FW	8.66 ± 2.09	8.13 ± 1.99	−0.53 ± 0.27	
	GK	10.99 ± 4.93	10.51 ± 4.01	−0.48 ± 0.94	
	Total	8.93 ± 2.51	8.37 ± 2.25	−0.56 ± 0.69	
Fat (kg) (Faulkner)	MF	10.14 ± 2.54	9.32 ± 2.49	−0.81 ± 0.30 *	
	DF	9.43 ± 1.38	9.20 ± 1.32	−0.23 ± 0.69	
	FW	9.89 ± 2.04	9.45 ± 1.95	−0.44 ± 0.31	0.021 *
	GK	11.51 ± 4.32	11.65 ± 4.02	0.15 ± 0.31	
	Total	9.93 ± 2.17	9.53 ± 2.12	−0.40 ± 0.54	
Fat (kg) (Durnin)	MF	13.86 ± 3.82	12.68 ± 3.99	−1.18 ± 0.61	
	DF	12.68 ± 2.25	12.41 ± 2.03	−0.27 ± 1.02	
	FW	13.31 ± 2.88	12.80 ± 2.96	−0.51 ± 0.51	0.036 *
	GK	16.00 ± 7.13	16.24 ± 6.16	0.25 ± 0.99	
	Total	13.45 ± 3.35	12.94 ± 3.29	−0.52 ± 0.86	
Fat (kg) (Jackson and Pollock)	MF	12.68 ± 4.44	10.94 ± 3.95	−1.74 ± 1.00 *	
	DF	10.69 ± 1.96	10.32 ± 1.94	−0.37 ± 1.06	
	FW	11.36 ± 2.93	10.46 ± 2.68	−0.90 ± 0.46	0.018 *
	GK	14.33 ± 6.68	13.72 ± 5.64	−0.61 ± 1.09	
	Total	11.69 ± 3.47	10.81 ± 3.09	−0.88 ± 1.00	

MF: midfielders; DF: defenders; FW: forwards; GK: goalkeepers. * Significant differences between positions in pre-season variation (*p* < 0.05).

**Table 6 nutrients-16-02799-t006:** Descriptive statistics at baseline and at the end of pre-season and variation of % of FM calculated with different equations adapted for female athletes: mean and standard deviations (mean ± SD).

Variable	Position	Baseline	Final	∆ Final—Baseline	*p*-Value
%Fat (Withers)	MF	18.27 ± 4.65	16.31 ± 4.90	−1.96 ± 0.76 *	0.003 *
	DF	16.62 ± 2.96	16.14 ± 2.80	−0.48 ± 1.12	
	FW	16.72 ± 3.07	15.97 ± 3.28	−0.75 ± 0.43	
	GK	18.91 ± 4.86	18.60 ± 3.84	−0.31 ± 1.03	
	Total	17.242 ± 3.5	16.34 ± 3.52	−0.90 ± 1.02	
%Fat (Slaughter)	MF	22.38 ± 6.26	19.93 ± 5.09	−2.46 ± 1.78 *	
	DF	19.10 ± 3.47	18.66 ± 3.57	−0.43 ± 0.69	
	FW	19.37 ± 4.14	18.62 ± 3.79	−0.75 ± 0.76	0.004 *
	GK	22.59 ± 6.90	21.55 ± 5.24	−1.04 ± 1.66	
	Total	20.26 ± 4.77	19.20 ± 4.08	−1.06 ± 1.35	
%Fat (Carter)	MF	16.40 ± 4.06	14.53 ± 3.57	−1.87 ± 0.88 *	0.002 *
	DF	14.38 ± 1.90	13.97 ± 1.90	−0.41 ± 0.90	
	FW	14.61 ± 2.38	13.77 ± 2.24	−0.85 ± 0.28	
	GK	16.54 ± 3.85	15.74 ± 3.01	−0.80 ± 0.89	
	Total	15.12 ± 2.86	14.19 ± 2.51	−0.93 ± 0.91	
%Fat (Faulkner)	MF	17.32 ± 2.86	16.09 ± 2.73	−1.23 ± 0.40 *	
	DF	16.54 ± 1.57	16.08 ± 1.36	−0.46 ± 0.85	
	FW	16.71 ± 1.91	16.03 ± 1.80	−0.68 ± 0.36	0.011 *
	GK	17.46 ± 2.65	17.51 ± 2.63	0.06 ± 0.13	
	Total	16.86 ± 2.05	16.19 ± 1.94	−0.67 ± 0.67	
% Fat (Durnin)	MF	23.65 ± 4.59	21.81 ± 4.91	−1.84 ± 0.76 *	
	DF	22.22 ± 2.97	21.67 ± 2.49	−0.54 ± 1.35	
	FW	22.48 ± 2.96	21.70 ± 3.26	−0.79 ± 0.89	0.020 *
	GK	24.06 ± 5.35	24.32 ± 4.50	0.27 ± 0.88	
	Total	22.80 ± 3.50	21.95 ± 3.49	−0.86 ± 1.18	
%Fat (Jackson and Pollock)	MF	21.54 ± 5.84	18.76 ± 5.15	−2.78 ± 1.34 *	
	DF	18.73 ± 2.63	18.01 ± 2.53	−0.71 ± 1.46	
	FW	19.13 ± 3.35	17.68 ± 2.95	−1.45 ± 0.61	0.006 *
	GK	21.52 ± 5.30	20.48 ± 4.45	−1.04 ± 1.07	
	Total	19.77 ± 4.03	18.30 ± 3.50	−1.47 ± 1.38	

MF: midfielders; DF: defenders; FW: forwards; GK: goalkeepers. * Significant differences between positions in pre-season variation (*p* < 0.05).

Moreover, Table 7 and Table 8 show the descriptive statistics at the beginning and end of pre-season as well as their evolution for absolute and relative values of MM, respectively. Focusing on the kg of muscle mass and playing position (Table 7), Poortmans’ equation shows statistically significant differences (*p* < 0.05) when comparing goalkeepers with defenders (*p* = 0.021) and forwards (*p* = 0.031). In contrast, Table 8 shows statistically significant differences in % muscle between midfielders versus defenders (*p* < 0.05) with Lee’s equation (*p* = 0.017). In summary, the only advantage observed for goalkeepers compared to other positions was an increase in muscle mass (using Poortmans’ equation) compared to defenders and forwards.

Table 9 shows the descriptive statistics at the beginning and end of pre-season as well as their evolution for somatotype. In turn, Figure 1 shows the somatotype profiles of female players by position at the baseline and the end of pre-season. From the 13 categories proposed by Heath and Carter’s method [33], two categories were observed. Initially, midfielders and forwards were classified as endomorphic mesomorphs (mesomorphy is dominant and endomorphy is greater than ectomorphy), while defenders and goalkeepers were mesomorph–endomorph (endomorphy and mesomorphy are equal or do not differ by more than 0.5, and ectomorphy is smaller). By the end of pre-season, it was observed that midfielders and forwards maintained their endomorphic mesomorph classification, while defenders changed from mesomorph–endomorph to endomorphic mesomorph and goalkeepers maintained their mesomorph–endomorph classification throughout pre-season. It should be noted that significant changes were observed mainly in the endomorphic component; specifically, the position that most decreased in this component were midfielders and significant decreases were observed versus defenders (*p* = 0.034) and goalkeepers (*p* = 0.023). In this research, it has been observed that midfielders, defenders, and forwards have a greater mesomorphy component than endomorphy, which differs from previous studies. Regarding midfielders, other studies have observed balanced endomorph [10,39], mesomorphic endomorph [10], and mesomorph–endomorph [36,38] somatotypes. For defenders in other studies, balanced endomorph [10], mesomorphic endomorph [10,38,39], and central [36] somatotypes have been observed. In regards to forwards, in other studies, balanced endomorph [39], mesomorph–endomorph [10,38], and central [36] somatotypes have been observed. As for goalkeepers, the somatotype observed in other studies is mesomorphic endomorph [10,38,39], which contrasts with the somatotype observed in the goalkeepers in this research study, whose endomorphy and mesomorphy are equal.

Table 10 shows the descriptive statistics at baseline and at the end of pre-season and the variation of BIA values. There were observed statistically significant differences (*p* < 0.05) in only three variables (relative and absolute values of FM and relative values of water). Regarding the values of FM, as well as the values obtained by anthropometry, a decrease in FM is observed in the entire population. Observing the evolution of each playing position, midfielders are the group with the greatest decrease in FM (both absolute and relative) compared with defenders and goalkeepers who increased FM. Considering the relative values between positions, there were differences between midfielders and defenders (*p* = 0.025) and between defenders and forwards (*p* = 0.040). In turn, absolute values showed a significant difference between midfielders and defenders (*p* = 0.017). Finally, the % of water significantly varied between midfielders and defenders (*p* = 0.009) and between defenders and forwards (*p* = 0.009).

In order to complete the approach of the body composition analysis, BIA analysis was used to assess different parameters of body composition, including bone mass. It is worth mentioning that although BIA results refer to the term “bone”, this can be misleading as it suggests the weight of the entire skeleton, whereas it actually estimates the bone mineral content (BMC) [51]. The BMC is the amount of minerals (calcium and phosphorus) expressed in grams. It should be emphasized that the BMC is not the same as the weight of the entire skeleton, which contains bone marrow, bone cells, connective tissue, cartilage tissue, water, blood vessels, and nerves, in addition to the mineral elements. The human skeleton represents 9–14% of the body weight of a thin person and the BMC is on average 40% of the skeletal weight [51]. Therefore, both fat-free mass (muscle, bone, tissue, water, and all other fat free mass in the body) and muscle mass (bone-free lean tissue mass) may be overestimated because skeletal weight is underestimated by assessing BMC rather than the actual skeletal weight.

**Table 10 nutrients-16-02799-t010:** Descriptive statistics at baseline and at the end of pre-season and variation of bioimpedance values: mean and standard deviations (mean ± SD).

Variable	Position	Baseline	Final	∆ Final—Baseline	*p*-Value
Fat mass %	MF	22.58 ± 6.09	21.03 ± 6.46	−1.55 ± 1.44 *	0.010 *
	DF	21.74 ± 4.04	22.32 ± 3.61	0.58 ± 1.70 *	
	FW	23.29 ± 4.72	22.06 ± 5.29	−1.24 ± 1.50	
	GK	25.27 ± 6.86	25.87 ± 5.73	0.60 ± 1.15	
	Total	22.75 ± 4.90	22.24 ± 5.02	−0.51 ± 1.77	
Fat mass (kg)	MF	13.39 ± 4.81	12.33 ± 4.87	−1.06 ± 0.92 *	
	DF	12.48 ± 3.18	12.86 ± 3.00	0.38 ± 1.07	
	FW	13.95 ± 4.28	13.29 ± 4.54	−0.73 ± 0.90	0.008 *
	GK	16.93 ± 8.25	17.37 ± 7.10	0.43 ± 1.16	
	Total	13.56 ± 4.41	13.25 ± 4.37	−0.32 ± 1.14	
Fat-free mass (kg)	MF	44.53 ± 2.85	44.70 ± 2.75	0.18 ± 0.97	0.501
	DF	44.34 ± 1.68	44.18 ± 1.80	−0.17 ± 1.01	
	FW	44.84 ± 2.64	45.28 ± 2.42	0.45 ± 1.01	
	GK	47.60 ± 5.72	47.97 ± 5.14	0.37 ± 0.61	
	Total	44.83 ± 2.75	44.99 ± 2.68	0.16 ± 0.97	
Muscle mass (bone-free lean tissue mass) (kg)	MF	42.26 ± 2.71	42.43 ± 2.62	0.16 ± 0.91	
	DF	42.09 ± 1.60	41.93 ± 1.71	−0.17 ± 0.96	
	FW	42.56 ± 2.51	42.98 ± 2.30	0.43 ± 0.98	0.490
	GK	45.20 ± 5.46	45.53 ± 4.91	0.33 ± 0.57	
	Total	42.56 ± 2.62	42.70 ± 2.55	0.15 ± 0.92	
Water (kg)	MF	31.15 ± 2.25	31.20 ± 2.20	0.05 ± 0.74	
	DF	31.11 ± 1.33	30.89 ± 1.45	−0.22 ± 0.80	
	FW	31.81 ± 2.11	32.13 ± 1.93	0.32 ± 0.73	0.360
	GK	33.90 ± 5.03	34.20 ± 4.59	0.30 ± 0.46	
	Total	31.59 ± 2.28	31.66 ± 2.25	0.07 ± 0.74	
Water %	MF	54.09 ± 3.76	55.08 ± 3.88	0.99 ± 0.97 *	
	DF	54.90 ± 2.67	54.30 ± 2.24	−0.60 ± 1.09 *	
	FW	54.40 ± 3.21	55.26 ± 3.46	0.86 ± 1.00	0.003 *
	GK	53.03 ± 3.55	52.70 ± 2.91	−0.33 ± 0.72	
	Total	54.38 ± 3.09	54.65 ± 3.09	0.27 ± 1.22	
Bone mineral content (kg)	MF	2.26 ± 0.15	2.28 ± 0.14	0.01 ± 0.06	
	DF	2.26 ± 0.08	2.25 ± 0.09	0.00 ± 0.06	
	FW	2.29 ± 0.16	2.313 ± 0.14	0.03 ± 0.05	0.781
	GK	2.25 ± 0.07	2.30 ± 0.00	0.05 ± 0.07	
	Total	2.27 ± 0.12	2.28 ± 0.11	0.09 ± 0.40	
Basal metabolic rate (kcal)	MF	1356.38 ± 90.04	1356.88 ± 91.77	0.50 ± 25.80	
	DF	1356.09 ± 55.90	1351.83 ± 60.40	−5.09 ± 26.68	
	FW	1380.63 ± 95.73	1389.75 ± 95.89	9.13 ± 27.30	0.768
	GK	1354.00 ± 2.83	1376.00 ± 4.24	22.00 ± 7.07	
	Total	1362.79 ± 74.57	1364.90 ± 76.82	47.53 ± 249.38	

MF: midfielders; DF: defenders; FW: forwards; GK: goalkeepers. * Significant differences between positions in pre-season variation (*p* < 0.05).

Table 11 presents the correlation coefficients between the BIA analysis (fat mass %, fat mass (kg), fat-free mass (kg), and muscle mass (bone-free lean tissue mass) (kg)) and those obtained through anthropometry. Regarding fat mass %, all correlations were statistically significant (*p* < 0.001). The highest correlation was observed with the Faulkner’s equation (*r* = 0.817; “very high” range of correlation). Subsequently, “very high” correlations were also observed with other equations, such as Durnin (*r* = 0.758), Jackson and Pollock (*r* = 0.746), and Carter (*r* = 0.709). Similarly, fat mass (kg) showed a strong (*p* < 0.001) correlation with all the equations, especially with the Faulkner’s equation (*r* = 0.953; “almost perfect” range of correlation) and the Durnin’s equation (*r* = 0.920; “almost perfect” range of correlation). Figure 2 shows the highest correlations between % and kg fat and BIA. The comparative analysis between anthropometric equations and BIA reveals that, although Faulkner’s equation demonstrates the strongest correlation, Durnin’s equation shows a greater similarity in the results obtained at both the beginning and the end of the study. This finding suggests that, despite the Faulkner equation potentially being influenced by various factors such as sample size and measurement precision, the consistency in the similarity of values between BIA and anthropometric equations, particularly with Durnin’s equation, is notable. It is important to note that the highest correlation can be observed with the two equations that do not consider lower body skinfolds. The finding about the high correlation between Faulkner and BIA is concordant with a previous study in professional male athletes [52]. For instance, previous research has demonstrated that anthropometric equations can be a viable alternative to BIA in estimating body composition in athletes [53,54,55], provided the specificities of the studied population are considered. However, it is important to note that most previous studies have been conducted on male or mixed populations, leaving a gap in the literature regarding the correlation of these equations in female populations, especially in specific sports like women’s soccer. Therefore, future studies should focus on validating these equations in diverse populations of female athletes and comparing them with more precise reference methods to strengthen the applicability of these tools in sports practice. Finally for fat-free mass (kg) and muscle mass (bone-free lean tissue mass) (kg), all correlations were statistically significant (*p* < 0.001). Lee’s equation had the highest correlation coefficients (*r* = 0.813 and *r* = 0.812, respectively; “very high” level of correlation in both cases). Since BIA measures BMC instead of bone mass directly, the values of fat-free mass and muscle mass (bone-free lean tissue mass) are overestimated. Due to this limitation, the results should be interpreted with caution.

## 4. Limitations

In this research, the analysis was limited to a sample of only 34 female players, which is a clear limitation. Therefore, further research with larger sample sizes is needed to confirm these findings. Another limitation was the number of goalkeepers, which means that perhaps the data from this population do not have more statistical power with respect to the rest of the positions, although it is true that as there were two teams, this problem was difficult to solve as there are only two players for this position in each team.

## 5. Conclusions

This study evaluated anthropometric and body composition changes during pre-season among Spanish professional female soccer players according to their playing positions. The results revealed that midfielders showed the most significant improvements in body composition, including reductions in skinfolds and fat mass. In contrast, defenders exhibited the least improvement, starting with lower baseline values and demonstrating minimal changes. Goalkeepers had distinctive characteristics, such as a larger wingspan and smaller ankle diameter, with minimal changes in upper body skinfolds, leading to no significant decrease in body fat percentage for the equations that include upper body measurements. Importantly, the study also demonstrated a high correlation between anthropometric measurements and BIA for estimating body composition, with the Faulkner and Durnin equations showing the strongest correlations. This reinforces the reliability of using both methods interchangeably in monitoring and assessing body composition changes. Overall, these findings highlight the positional differences in body composition changes during pre-season and suggest that tailored training and nutrition programs could optimize performance for each position.

## Figures and Tables

**Figure 1 nutrients-16-02799-f001:**
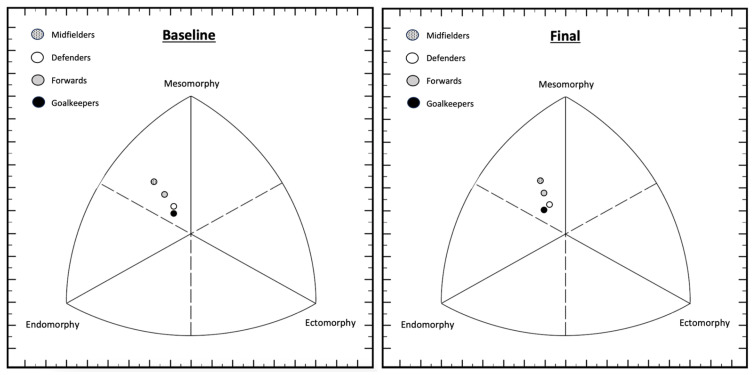
Somatotype of female players by position at the baseline and the end of pre-season.

**Figure 2 nutrients-16-02799-f002:**
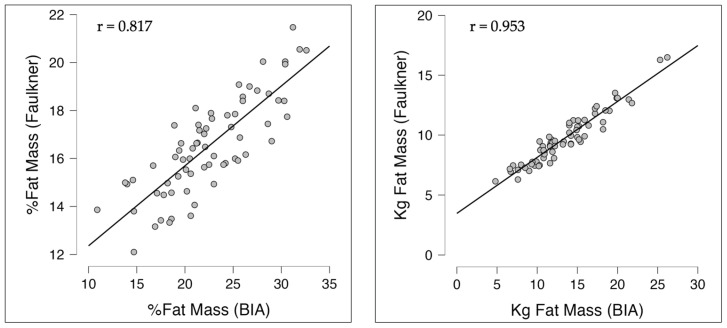
Graphs representing the correlation between the fat mass values obtained with the Faulkner anthropometric equation and the values obtained with BIA.

**Table 1 nutrients-16-02799-t001:** Demographic data of the players at baseline: mean and standard deviations (mean ± SD).

	Mean ± SD
n	34
Gender	Female
Age	23.06 ± 4.29
Height (cm)	164.15 ± 5.84
Weight (kg)	58.39 ± 6.62
BMI (kg/m^2^)	21.64 ± 1.77
∑6 Skinfolds (mm)	74.57 ± 18.48

∑6 Skinfolds (triceps, subscapular, supraspinal, abdominal, front thigh, and medial calf).

**Table 7 nutrients-16-02799-t007:** Descriptive statistics at baseline and at the end of pre-season and variation of kg of MM calculated with different equations adapted for female athletes: mean and standard deviations (mean ± SD).

Variable	Position	Baseline	Final	∆ Final—Baseline	*p*-Value
Muscle (kg) (Poortmans)	MF	23.00 ± 2.10	23.56 ± 2.24	0.55 ± 0.60	0.024 *
	DF	22.93 ± 2.54	23.19 ± 2.51	0.26 ± 0.46 *	
	FW	23.94 ± 3.09	24.24 ± 3.08	0.30 ± 0.41 *	
	GK	23.25 ± 2.70	24.48 ± 2.47	1.24 ± 0.58	
	Total	23.30 ± 2.58	23.73 ± 2.58	0.43 ± 0.54	
Muscle (kg) (Lee)	MF	22.77 ± 1.90	23.21 ± 1.98	0.44 ± 0.56	
	DF	22.59 ± 2.00	22.69 ± 1.94	0.11 ± 0.54	
	FW	23.60 ± 1.84	23.81 ± 1.83	0.20 ± 0.40	0.110
	GK	23.91 ± 2.60	24.76 ± 2.36	0.85 ± 0.40	
	Total	23.08 ± 1.95	23.36 ± 1.96	0.28 ± 0.52	

MF: midfielders; DF: defenders; FW: forwards; GK: goalkeepers. * Significant differences between positions in pre-season variation (*p* < 0.05).

**Table 8 nutrients-16-02799-t008:** Descriptive statistics at baseline and at the end of pre-season and variation of % of MM calculated with different equations adapted for female athletes: mean and standard deviations (mean ± SD).

Variable	Position	Baseline	Final	∆ Final—Baseline	*p*-Value
%Muscle (Poortmans)	MF	39.94 ± 3.58	41.48 ± 4.14	1.55 ± 1.66	0.083
	DF	40.33 ± 2.97	40.68 ± 3.39	0.35 ± 0.63	
	FW	40.87 ± 4.23	41.58 ± 4.16	0.71 ± 0.72	
	GK	36.57 ± 4.48	37.86 ± 3.44	1.29 ± 1.18	
	Total	40.08 ± 3.69	40.91 ± 3.81	0.83 ± 1.09	
%Muscle (Lee)	MF	39.51 ± 3.00	40.81 ± 2.78	1.30 ± 1.16 *	
	DF	39.76 ± 2.10	39.82 ± 2.22	0.06 ± 0.86	
	FW	40.41 ± 3.43	40.94 ± 3.33	0.54 ± 0.54	0.030 *
	GK	37.59 ± 3.95	38.30 ± 3.28	0.71 ± 0.80	
	Total	39.72 ± 2.91	40.28 ± 2.82	0.56 ± 0.94	

MF: midfielders; DF: defenders; FW: forwards; GK: goalkeepers. * Significant differences between positions in pre-season variation (*p* < 0.05).

**Table 9 nutrients-16-02799-t009:** Descriptive statistics at baseline and at the end of pre-season and variation of somatotype values: mean and standard deviations (mean ± SD).

Variable	Position	Baseline	Final	∆ Final—Baseline	*p*-Value
Endomorphy	MF	3.87 ± 1.00	3.44 ± 0.97	−0.43 ± 0.15 *	0.012 *
	DF	3.53 ± 0.66	3.39 ± 0.53	−0.14 ± 0.31	
	FW	3.55 ± 0.62	3.39 ± 0.65	−0.18 ± 0.14	
	GK	3.70 ± 1.00	3.72 ± 0.82	0.02 ± 0.17	
	Total	3.63 ± 0.75	3.43 ± 0.69	−0.21 ± 0.25	
Mesomorphy	MF	4.84 ± 0.98	4.74 ± 0.95	−0.11 ± 0.15	
	DF	4.04 ± 0.68	4.03 ± 0.75	−0.004 ± 0.18	
	FW	4.31 ± 0.97	4.32 ± 0.86	0.01 ± 0.18	0.479
	GK	4.01 ± 0.77	4.05 ± 0.47	0.04 ± 0.30	
	Total	4.31 ± 0.89	4.29 ± 0.83	−0.02 ± 0.18	
Ectomorphy	MF	2.13 ± 0.87	2.26 ± 0.93	0.14 ± 0.26	0.159
	DF	2.71 ± 1.00	2.66 ± 0.99	−0.05 ± 0.24	
	FW	2.31 ± 0.81	2.36 ± 0.87	0.05 ± 0.17	
	GK	2.89 ± 0.64	2.72 ± 0.35	−0.17 ± 0.29	
	Total	2.46 ± 0.89	2.48 ± 0.88	0.02 ± 0.23	

MF: midfielders; DF: defenders; FW: forwards; GK: goalkeepers. * Significant differences between positions in pre-season variation (*p* < 0.05).

**Table 11 nutrients-16-02799-t011:** Correlation between bioimpedance and anthropometry.

Bioimpedance vs. Anthropometry	Pearson’s *r*	Lower 95% CI	Upper 95% CI	*p*-Value
% Fat Mass				
% Withers	0.698	0.551	0.803	<0.001
% Slaughter	0.583	0.400	0.721	<0.001
% Carter	0.709	0.567	0.811	<0.001
% Faulkner	0.817	0.718	0.883	<0.001
% Durnin	0.758	0.635	0.844	<0.001
% Jackson and Pollock	0.746	0.618	0.836	<0.001
Fat Mass (kg)				
Kg Withers	0.871	0.798	0.919	<0.001
Kg Slaughter	0.807	0.704	0.877	<0.001
Kg Carter	0.885	0.820	0.928	<0.001
Kg Faulkner	0.953	0.924	0.971	<0.001
Kg Durnin	0.920	0.873	0.950	<0.001
Kg Jackson and Pollock	0.895	0.835	0.934	<0.001
Fat-Free Mass (kg)				
Kg Poortmans	0.657	0.497	0.774	<0.001
Kg Lee	0.813	0.712	0.880	<0.001
Muscle Mass (Bone-free lean tissue mass) (kg)				
Kg Poortmans	0.656	0.495	0.774	<0.001
Kg Lee	0.812	0.711	0.880	<0.001

*r* = 0.0–0.09 were considered trivial, *r* = 0.10–0.29 small, *r* = 0.30–0.49 moderate, *r* = 0.50–0.69 high, *r* = 0.70–0.89 very high, *r* = 0.90–0.99 almost perfect, and *r* = 1 perfect correlation [56].

## Data Availability

The data are contained within the article.

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
