# Peer review of "Anthropometric and Body Composition Changes during Pre-Season of Spanish Professional Female Soccer Players According to Playing Position"

_nutrients, 2024, doi:10.3390/nu16162799_

Round 1
Reviewer 1 Report
Comments and Suggestions for Authors
Dear authors,
I am writing to express my sincere gratitude for the opportunity to review the manuscript entitled “Anthropometric and Body Composition Changes During Pre-season of Spanish Professional Female Soccer Players According to Playing Position” submitted to Nutrients.
This manuscript is within the scope of Nutrient, is well conducted, is well written, and also is very interesting in its context.
Please, consider the following minor comments:
- In order to provide a broader context, you must indicate the growth rate of the number of clubs or licences both in Europe and in Spain (lines 38 and 41).
- A few days ago, MDPI published a new research focus on professional female footballers (https://doi.org/10.3390/app14146349) . Since, as you state in the introduction, there is a lack of articles on this topic, and in order to make your study more up-to-date, I recommend that you include it.
- Explain what you mean by "the correlation of anthropometric equations with bioimpedance (BIA)". These concepts must be described, as they are not mentioned in the preceding lines. If this is a secondary aim, it must be stated.
- Was the normality of the data analysed?
- In the methodology section, the authors state that 8 folds were measured, but only 6 folds are reported in Table 1. Add a footnote indicating which folds are reported.
- Change from p < 0.05 to p ≤ 0.05 in line 177.
- If you report an exact p-value you should indicate p = value and not < or >. Check lines 181-183 and the hole document.
- Table 2 would be easier to read if the game positions were put on the same line and below the values of each variable.
- In table 11, check for typing errors.
- Figure 2 must be deleted (data already included in table 11)
- Add a short paragraph with the limitation of your study and its practical implications.
Comments on the Quality of English Language
The quality of the English is good.
Reviewer 2 Report
Comments and Suggestions for Authors
The study provides a lot of interesting anthropometric data for elite Spanish female soccer players, however, the observed set is unfortunately small (n= 34) and the comparison between 8 midfielders, 11 forwards, 12 defenders and 3 goalkeepers has only indicative value and cannot be considered representative for female football players. An appropriate sample of participants for the monitored data could be determined by power analysis.
An interesting comparison of the anthropometric profile with the results of foreign studies also has limited informative value, because ethnic differences in anthropometric characteristics can significantly influence the compared values.
The authors focused on the comparison of anthropometric characteristics before and after completing the preparatory period and found favorable changes in body composition and selected body dimensions. The authors also compared the results of the bioimpedance examination and a number of calibration methods and recommended the two most suitable methods of skinfold calibration for the practice of women's football
Comments on the manuscript:
Chapter 3 "Results" should be labeled "Results and Discussion" because it deals with a detailed discussion of the results in addition to the presentation of the results.
The manuscript should be supplemented with a subsection Limitations of the study, where the authors would state the strengths and weaknesses of their study.
In Table 1, the average values and SD for BMI are given as 21.64 +/- 21.59 kg/m2, the SD value should be corrected.
On line #220, the word "payers" should be corrected to "players".
Reviewer 3 Report
Comments and Suggestions for Authors
Thanks to give me the opportunity to review this article.
At row 250, refering to relaxed arm mesurments, the authors wrote:"Comparing playing positions, goalkeepers and midfielders were the only that showed statistically significant differences (p < 0.017)." but in table 3 seems that only midfielders value is statistical significant.
Over all the article is good written and I like the experiment design.
I think readers it could be appratiated if the authors explain the choices of the anthropometric parameters.
Finally, I think is better to write the limitations of the article and write something more then a simple "suggest that tailored training and nutrition programs"
Comments on the Quality of English Language
very good
